# An Exploration of the Impact of COVID-19 on Police Demand, Capacity and Capability

Eric Halford 

Rabdan Academy, Abu Dhabi P.O. Box 114646, United Arab Emirates; ehalford@ra.ac.ae

**Abstract:** This project report outlines the findings of an initial exploratory study of the impact of the coronavirus (COVID-19) pandemic on the demand, capacity, and capability of the police within the United Kingdom. Freedom of information requests provide data regarding employees affected by COVID-19, including those working from home. A survey of police employees adds richness by exploring the departments and specialist capabilities affected. Preliminary results indicate a shift in demand away from property-related and violent crime, to online criminality, and disorders such as anti-social behavior and breaches of coronavirus legislation. Combined with high volumes of absent employees throughout 2020, the study postulates a reduction in police satisfaction, trust, and confidence may have occurred in the response to cyber investigation and policing anti-social behavior. Investment in agile technology to increase workforce flexibility and improved contingency planning are identified as requirements to prepare for future pandemics and avoid repetition.

**Keywords:** policing; police; crime; COVID-19; coronavirus; pandemic

## 1. Introduction

Policing was right at the frontline of the pandemic. The introduction of new legislation provided under the United Kingdom's (UK) coronavirus act included new laws for the police to issue fixed penalty notices (FPNs) for a variety of legislative breaches, a practice that was introduced widely around the world. As a result of stay-at-home orders, social distancing, and the new legislation, the demand facing the police service is likely to have altered in ways that are both predictable and unforeseen. It is natural to presume that crimes that rely on the interaction of people, such as violent and sexual offences, or those that require the presence of a capable guardian to prevent, such as burglary, were likely to reduce as a result of conditions that significantly restrict mobility. In addition, certain crimes require access to allow an offender opportunity to commit them, shoplifting being the most obvious. As a result, laws closing down much of the shopping industry were likely to have reduced such offending. These predictions held true when subjected to early research (Halford et al. 2020). However, since initial studies emerged, despite widespread international research there has only been a limited number of studies within the UK that have continued to investigate the impact of COVID-19 on policing (Buil-Gil et al. 2020; CSEW 2021; Dixon et al. 2020; Langton et al. 2020; Neanidis and Rana 2021; Nivette et al. 2021). Studies conducted have focused on the demand changes experienced by the police, predominately by analyzing recorded crimes.

There is an absence, however, of any literature in the UK, or globally, that has explored the impact of COVID-19 on policing in the context of its effect on available capacity and capabilities. This is important because demand, capacity, and capability are three interconnected issues. Understanding demand and how this changed during the pandemic is invaluable as this enables police services to better prepare how they respond. Vital to their response is also the understanding of the level of available capacity, as defined by the numbers of police officers and civilian police staff, as any failure to adequately match incoming demand with the requisite level of capacity is likely to result in a significant

impact on service delivery. In addition, capability, as defined by the specialist skills and training that certain police officers and civilian staff possess, is also key as certain forms of incoming demand can only be managed by officers or staff with specific qualifications. When considering these factors, we can begin to appreciate how each has a direct and significant influence over the other. To balance these three vital elements and achieve, and maintain, an equilibrium within the police service so that they can meet service requirements is an essential balancing act. Any dramatic fluctuation or rapid change in any of these three components can create an underlying demand/capacity/capability mismatch (Walley and Adams 2019) which if not responded to effectively can undermine the police's ability to meet their core requirements. If this occurs then there is a genuine possibility that this will negatively affect the satisfaction of the public with the police response and, as such, reduce their confidence and trust in the service. Any reduction in these factors is likely to undermine the legitimacy of the police.

To begin understanding such issues this project makes the first attempt to understand the impact of COVID-19 on the makeup of demand for the police service in the UK as part of a wider systematic research program examining the impact of COVID-19 on policing. In doing so it begins to fill an important research gap whilst simultaneously identifying data and methodological restrictions relating to survey participants that subsequent, more in-depth studies can use to inform their preparation and collaboration efforts to improve the validity of future work in this area by probing its associated findings. It does this by providing a '*bird's eye view*' of the research to date on policing demand in the UK. To complement this, the study furthers understanding by examining the absence levels of police employees throughout the pandemic and any associated impact on access to key police capabilities. As a result of this analysis, a discussion is then framed in the context of the overall effect of COVID-19 on policing, demand, capacity, and capability before providing potential solutions for services to better prepare themselves for further pandemics.

## 2. Literature Review of Police Demand, Capacity and Capability

Prior to 2017, the literature on police demand that helped explore the required levels of capacity and capability was extremely limited and focused almost entirely on calls for service (Adler et al. 2013; Brooks et al. 2011; Curtin et al. 2007; D'Amico et al. 2002; De La Cruz 2016; Decker et al. 2007; Fleming and Grabosky 2009; Greasley and Smith 2017; Green 1984; Groff et al. 2014; Heller and Markland 1970; Johnson and Rhodes 2009; Maxfield 1982; Mazzerolle et al. 2002; Moore and Braga 2003; Sacks 2003; Sarac et al. 1999; Taylor and Huxley 1989; Taylor Griffiths et al. 2015; Walley 2013; Zaki et al. 1997; Zhang and Brown 2013). However, since that time there has been a diversification in the studies conducted that examine the nature and context of the demands placed upon the police, particularly within the United Kingdom (UK). This was driven by a 2015 report published by Her Majesty's Inspectorate of Constabularies, Fire and Rescue Service (HMICFRS). This outlined a necessity for the police to have a greater understanding of the demand they faced both at the time, and in the future, so that they may better adapt to the changing environment facing policing (Loveday 2017). The inspectorate was very clear, stating; "*the lack of understanding of future demand was adversely affecting how forces were planning for the future*" (HMICFRS 2015).

Her Majesty's Inspectorate of Constabularies (HMIC 2008, p. 16) has previously defined capacity and capability, stating; "capability refers to the skills and expertise the force can summon up. Its 'capacity' is the strength and depth of its resources". HMICFRS has suggested that police services in the UK have a strong understanding of their capacity but they often have very little awareness of how much was required to meet the demand the service faced, and importantly, the capabilities of the officers, police community support officers (PCSOs), and police staff (unsworn employees) required to achieve it (HMICFRS 2015). In an effort to reach a more informed position, in 2017 the UK National Police Chiefs Council (NPCC 2017) produced the report 'Better Understanding Demand: Policing the Future'. This detailed report contained an in-depth examination of all areas of police

demand and was subsequently underpinned by an academic scoping review on the subject (Laufs et al. 2020). Categories of police demand were devised including reactive demand, protective demand, which is often described as 'pro-active policing', and organizational demand, which is required to keep the system of policing functions. Finally, failure demand was outlined, which is any form of demand that was managed ineffectively and as a result, led to further demand generation (Laufs et al. 2020; NPCC 2017; Walley and Jennison-Phillips 2018b). Failure demand also includes work conducted that does not provide the customer, which, in a policing context, would generally be the victim of crime, an outcome that provided value to them (Benington and Moore 2011). Drivers of demand were described as a combination of temporal phases such as cyclical events, the seasons, vulnerability, localized issues, and baseline, 'every day, business as usual' demand (Laufs et al. 2020; NPCC 2017).

In addition to understanding the nature and complexity of police demand, other literature has sought to examine the responses to it (Fleming and Grabosky 2009; Walley and Adams 2019; Walley and Jennison-Phillips 2018a; Walley and Jennison-Phillips 2018b). Fleming and Grabosky (2009) have outlined the concept of rationing capacity and capabilities as a method used to respond to demand, and Walley and Adams (2019) subsequently confirmed its use by UK police services in operational environments. Other less frequently used approaches include the adjustment of risk thresholds (Walley and Adams 2019) and the use of crime harm measures (Laufs et al. 2020). The understanding of demand and the variety of responses to it are fundamental because they underpin how, when, and where police capacity is used, and which capability is deployed.

Despite its importance, there is very limited research that examines the area of capacity and capability within policing. The literature has examined the per capita approach where officer numbers are aligned with population levels (Adams 1994; Orrick 2008), the minimum staffing approach (Demers et al. 2007) which relates to internal policies that mandate the number of staff required to be on duty at a given time and date, the budgeted approach to capacity (Wilson et al. 2010), and finally, the workload approach, which is the most commonly cited within the literature and is considered amongst practitioners to be the most accurate way to create an equilibrium between demand and capacity (Hale 1994; Lumb 1996; Orrick 2008; Shane 2007; Wilson and McLaren 1977; Wilson and Weiss 2014). This study is unique within the literature as it cuts across all three areas (Demand, Capacity, and Capability) and is one of the first to explore how capacity and capability are driven by societal crises, namely the COVID-19 pandemic. In doing so, it contributes significantly to an area of literature that is limited in-depth and, as a result, holds a high level of public value. It also enables the police service in the UK to ensure it maintains the informed position advocated by the inspectorate in the face of unprecedented crises.

## 3. Literature on the Impact of COVID-19 on Police Demand in the United Kingdom

To further understand the value of this study it is important to briefly outline the literature regarding the impact of COVID-19 on policing within the UK. An examination of this literature has identified that the present studies available have focused on a single dominant form of police demand, namely reactive demand, and more specifically, that which is related to recorded crime rates (Buil-Gil et al. 2020; CSEW 2021; Dixon et al. 2020; Halford et al. 2020; Langton et al. 2020; Neanidis and Rana 2021; Nivette et al. 2021). This leaves the areas of protective, organizational, and failure demand absent of detailed empirical scrutiny with the exception of a single study (Maskály et al. 2021). This study, conducted using a survey of officers, does not distinguish the impact between the regions examined (United States, UK, and Europe), so although it provides an early indication, it falls short of the assessment required to provide meaningful information to underpin strategic, tactical, or operational policy decision-making. This review has also identified that the UK has only focused on recorded crime rates. This is a high proportion of reactive demand, but it is distinct from other forms such as calls for service, which have been examined in detail in other countries (Ashby 2020; Bullinger et al. 2021; Dai et al. 2021;

Koziarski 2021; Lum et al. 2020; Nix and Richards 2021; Richards et al. 2021) but not in the United Kingdom.

Table A1 (situated within the Appendix A) provides an overview of the research exploring how COVID-19 has impacted police demand that this review has identified. This overview shows that property crimes (theft, robbery, burglary, shoplifting) that relate to the routine activities, mobility, and interactions of the general population have been impacted the most. Within policing, particularly in the UK, these areas of crime are often responded to and investigated by level 1 professionalizing investigation program (PIP1) accredited constables. PIP1 is the introductory level of investigation training and is given to frontline uniformed officers. Violent offences, including homicide, domestic, and sexual violence have also been reduced. Frontline uniformed constables are also the first responders to these crimes. In contrast to property crime, most violent offences are subsequently investigated by officers in investigative roles such as those within what are known as police custody reception teams (CRTs) and criminal investigation departments (CID). With homicide being the most serious of all offences, this is investigated and managed by specialist and senior accredited detectives who are PIP2 and PIP3 qualified. We can also see that a category of offences and behaviors that would commonly be considered community-orientated problems (public order and criminal damage) is also reduced by a substantial amount. Conversely, anti-social behavior (ASB), often one of the policing priorities for community policing teams, is the area identified as having increased by the largest degree. Similarly, both organized crime and cyber-related offences both suffer significant increases.

These two crime forms usually attract a very specific policing capacity and capability. For instance, regardless of the nation, serious cyber-crimes usually require the use of trained digital media investigators and digital forensic technicians who are both experts trained in highly technologically skilled techniques. Similarly, combating organized crime often requires the use of pro-active capacity, which is drawn from intelligence, surveillance, or targeting departments. Additionally, in an unprecedented move, the role of the police was substantially adapted as they bore the main responsibility for the implementation of the new coronavirus legislation. As a result, they were used extensively to manage both reactive and proactive demand that related to breaches of government laws regarding stay-at-home orders, infringements of movement, and non-essential travel. This can be seen in the volume of fixed penalty notices (FPNs) issued, with every single FPN issued representing an impact on both the capacity and capability of the police that did not exist prior to the COVID-19 pandemic.

## 4. Importance of the Study

When we consider the changes in policing demand in the context of the capability required to respond, we can begin to understand how COVID-19 affected policing as a whole and begin to appreciate that this was in contrasting and nuanced ways. Within policing the maintenance of consent is fundamental to the trust and confidence of the public, and the ability of the police to legitimately deliver services (Jackson and Bradford 2010; Merry et al. 2012; Rix et al. 2009; Schaap 2020). The impact on the legitimacy of ineffective and inaccurate demand awareness and the interlinked capacity and capability planning is significant and it is unwise to understate this. For instance, research indicates that there is a high likelihood that if capacity and capability do not meet demand then the trust and confidence in the police could be negatively impacted as a result of the reduced ability of the police to deliver services effectively (Walley and Jennison-Phillips 2018a). Because of reductions in trust and confidence police legitimacy also reduces, and it is suggested that this can directly affect crime rates, causing them to increase (Cook 2015; Walley and Jennison-Phillips 2018a) as ordinary people become less inclined to abide by the law of the land. Specifically, Cook suggests that the capacity of the police has a direct impact on their ability to tackle certain crime forms, especially property-related crime (Cook 2015), and Walley and Jennison-Phillips (2018a) suggest that a drop in effectiveness would also undermine the police's ability to reduce re-offending and control social disorder.

In addition to exploring the potential impact of fluctuations in police demand, capacity, and capabilities on the area of legitimacy, the study is also important for practical reasons. For example, it may provide insights into other impacts such as those related to public safety and well-being, appropriate training and stress mitigation for officers, funding allocation, and staff recruitment to name a few.

## 5. Research Aims

The main purpose of this study was to help the police service in the United Kingdom understand the impact of COVID-19 on their ability to deliver services effectively and understand if, and where, any shortcomings may have occurred. Understanding this enables the police service to consider how it may respond to future major shifts in demand, capacity, and capability such as a future pandemic. The study initially aimed to achieve this by focusing on four specific questions: (1) what was the impact on police demand, (2) what was the impact on police capacity and capability, (3) what was the likely impact of any identified changes, and (4) how can the police service better respond to future pandemics? The study quickly identified that key data vital to answering some of these questions, specifically that relating to capacity and capability which requires data for recorded absences of police officers and staff, was not publicly available or reported on. This was especially true for data relating to COVID-19-related absences. As such, novel but restrictive data collection methods were required to be used. In addition, attrition experienced during the survey process also resulted in lower than anticipated participation. As a result, the ambition of the study was reassessed and although the questions posed are still considered, the findings are presented as a project report and not a full research article, with the intention of generating a call to arms for further research in this important area.

## 6. Methodology

To answer these questions, a mixed methodology has been used. First, the collation of data from freedom of information requests (FOIs) from police services in the United Kingdom is used to gather data regarding the impact on capacity and capability. The Freedom of Information Act 2000 provides public access to information held by public authorities, including police services. The Act covers any recorded information that is held in England, Wales, and Northern Ireland, with a legal obligation to reply unless specific criteria are present. As alluded to in the previous section, this method was used as there was no openly accessible information regarding the impact of COVID-19 on the capacity of the police service and as such, it was the only route available to access these data. FOIs were submitted to all UK police services requesting data that outlined the number of police officers and staff absent at any stage due to contraction of COVID-19, the requirement to self-isolate or shield at home. In addition, data for the number of staff that were able to work from home whilst absent were also requested.

The second method used was a survey of police employees in the UK. This method was chosen as it sought to understand the effect on operational access to capacity, and specialist skills and expertise (capabilities) for employees working within UK police services during the pandemic. This enabled an opportunity to get richer detail to support the FOI data with factors that may have affected the available capacity and capability, such as access to personal protective equipment. The study used a survey as it enabled a simple and effective way to enable *"the collection of information from a sample of individuals through their responses to questions"* (Check and Schutt 2011). To create the survey, 19 questions were split into 4 key categories. These included (1) demographic information, such as length of service and age (2) capability, which included details of the participant's present role and any specialist qualifications they held (3) capacity, such as periods of sickness or absence due to COVID-19 and (4) COVID related information which explored the circumstances of the diseases impact on the participant's ability to conduct their role, or other work effectively. The survey questions were then uploaded to a third-party web hosting service which

enabled digital completion of the survey by participants who could do so by following a link provided. The survey was voluntary, and this was declared at the outset.

It is important when conducting surveys to make sure that the population of interest is effectively targeted to identify suitable respondents (Ponto 2015). To achieve this, employees were accessed via the UK evidence based policing (EBP) champion network, which is coordinated by the UK College of Policing (CoP). These champions were accessed via the evidence-based champions coordinator in the CoP who shared the survey amongst police employees in the UK. Although preferable to access the entire target population, this is not always possible, and this was the case in this study and attrition occurred at various gateways, not least of which was the discretion of the EBP champion about whether the survey was shared, who with, and to what extent. Because of the use of this circulation approach, fewer participants engaged than would have with a nationally supported or sanctioned survey conducted in collaboration with the UK Police Federation or National Police Chiefs Council, for example. The conduct of interviews was considered, but it was felt that this approach was not suitable during COVID-19.

To display the findings, descriptive approaches are utilized. This is because amongst the primary target audiences for this article are practitioners, senior police leaders, and policy-makers. It has been suggested that the use of descriptive approaches is more effective than improving understanding (Conner and Johnson 2017) and therefore has a greater potential for impact.

### 7. Results

*7.1. Freedom of Information Requests*

Naturally, both police officers and staff were considered essential key workers throughout the period of the COVID-19 pandemic. Unlike other essential workers, the police were right at the frontline of the fight against the virus through their work with the public to engage, educate, encourage, and enforce coronavirus legislation. Similar to other emergency services around the world, the police could not simply work from home or cease engagement with people within communities, as by its nature, protecting communities is a *'hands on'* job. Unsurprisingly, the COVID-19 virus affected the police service in significant ways and this study seeks to understand the extent of the impact. To achieve this, all 43 regional UK police services, Police Scotland, and the Ministry of Defense (MoD) police were sent the same FOI request via email, totaling 45 police services. Table 1 shows the questions asked within the FOI request and their relative response rates and the mean and median levels of affected employees (both police officers and civilian staff members).

In total, 25 police services (58%) responded positively to the FOI inquiry. On the face of it, this may appear a low response level. However, it should be recognized that these data are not released routinely by police services due to their potentially sensitive nature and, as such, the level of response is a minor achievement in itself. This is evidenced by the fact that a number of services did not provide answers to some of the questions posed, and, as a result, the numbers that were meaningfully analysed had to be reduced from 16 to 8 (questions 3, 6, 7, 8, 11, 14, 15, and 16). To help further understand the impact on police capacity and capability, the FOI results are underpinned by the survey of police officers and staff.

Regardless of the reduction in questions analysed from the FOI responses a number of questions can still be posed of the data to support the research aims. This includes the volume of officers and staff that reported absent because of being infected by COVID-19, those absent due to having to self-isolate after contact with a confirmed or suspected COVID-19 case, and those required to shield at home to protect them from infection. The responses also enabled the study to explore how many of the absent employees were able to conduct work from home. A significant point that requires acknowledgement is that the figures reported do not provide the total number of days but only the number of staff absent for the specified reasons. This is important, as the length of the absences are case dependent. For example, officer A in one police service who is reported as absent due to

shielding may have done so for 31 days of a calendar month, whereas officer B in another service may have only shielded for 20 days; however, both are reported as a single shielding absence under the relevant month. Due to this fact, the deductions drawn should only be considered an indicative analysis. A much more comprehensive investigation would be required to fully understand the daily effect on capacity and capability. Regardless of the limitations, the study is still successful in providing important findings.

**Table 1.** List of Freedom of Information Request Questions and Response Rates.

| Freedom of Information Request Question | Responding Services (%) | Median No. of Employees | Mean (Average) No. of Employees |
|---|---|---|---|
| Q1: How many Police Officers reported absent as a result of contracting COVID-19 OFF Duty | 0 (0%) | Not Known | Not Known |
| Q2: How many reported absent as a result of contracting COVID-19 ON Duty | 0 (0%) | Not Known | Not Known |
| Q3: In TOTAL, both OFF and ON duty, how many reported absent due to contracting COVID -19 | 23 (51%) | 196 | 950 |
| Q4: How many reported absent as a result of having to self-isolate after being in contact with someone OFF duty who contracted COVID-19 | 1 (2.2%) | 797 | 797 |
| Q5: How many reported absent as a result of having to self-isolate after being in contact with someone ON duty who contracted COVID-19 | 1 (2.2%) | 197 | 197 |
| Q6: In TOTAL, how many reported absent due to having to self-isolate after being in contact with someone ON or OFF duty who contracted COVID-19 | 23 (51%) | 937 | 1309 |
| Q7: How many were absent having to shield from the risk of contracting COVID-19 in work | 20 (44%) | 93 | 171 |
| Q8: Of those absent due to self-isolation or shielding how many were able to conduct meaningful work from home | 17 (38%) | 639.5 | 841 |
| Q9: How many Police Staff reported absent as a result of contracting COVID-19 OFF Duty | 0 (0%) | Not Known | Not Known |
| Q10: How many reported absent as a result of contracting COVID-19 ON Duty | 0 (0%) | Not Known | Not Known |
| Q11: In TOTAL, both OFF and ON duty, how many reported absent due to contracting COVID | 24 (53%) | 122 | 522 |
| Q12: How many reported absent as a result of having to self-isolate after being in contact with someone OFF duty who contracted COVID-19 | 2 (4.4%) | 522 | 261 |
| Q13: How many reported absent as a result of having to self-isolate after being in contact with someone ON duty who contracted COVID-19 | 1 (2.2 %) | 55 | 55 |
| Q14: In TOTAL, how many reported absent due to having to self-isolate after being in contact with someone ON or OFF duty who contracted COVID-19 | 22 (49%) | 536 | 718 |
| Q15: How many were absent due to having to shield from risk of contracting COVID-19 in the workplace | 19 (42%) | 99 | 197 |
| Q16: Of those absent due to self-isolation or shielding how many were able to conduct meaningful work from home | 17 (38%) | 310 | 484 |

First, we can begin to understand the amount of available capacity that was impacted by COVID-19. Figure 1 shows the proportion of police officers absent on a month-by-month basis due to the three reasons explored in the study (infection, self-isolation, and

shielding). This analysis was conducted by accessing data regarding the number of serving police officers in the UK for those services that provided suitable data regarding affected officers in their FOI data return. By comparing overall available officer numbers from these police services (which provides a level of maximum available police officer capacity) against the total reported absences from all responding services for each month we can identify a proportionate effect. This analysis could not be repeated for police staff due to an absence of data regarding their numbers being publicly available. This process indicates that broadly speaking, the trends in the volume of absences for police officers reporting absent due to contracting COVID-19 or for self-isolation purposes, matches the peaks of the virus infection rate in the general populous in the UK (Bhatia et al. 2021). In wave 1 approximately 10% of all police officers were absent at some stage of April, for one of the stated reasons. In wave 2, this increased slightly during November 2020 before reaching the maximum level of absence in January 2021.

At that point, the analysis indicates that over 20% of police officers reported absent during the month of January 2021. A point worth noting here is that these findings include only COVID-19-related absences and not existing absences for other reasons such as an injury at work, bereavement, or other causes of illness, so the true figure was likely higher. In Figures 1 and 2 we illustrate the trends identified by the analysis conducted on individual police services. This is done to show the proportional absences vs. the overall available police capacity and is achieved by obtaining the number of serving officers for each police service and comparing these with the reported figures of police officers who were absent for the examined reasons (infection, self-isolation, and shielding) along with details of those able to work from home for each service. From this analysis, we can see that there appears to be a trend related to the size of the police service and the number of reported absences. It appears that smaller forces suffer a much greater proportional volume in reduced available police officers with the impact of self-isolation being the primary reason for this.

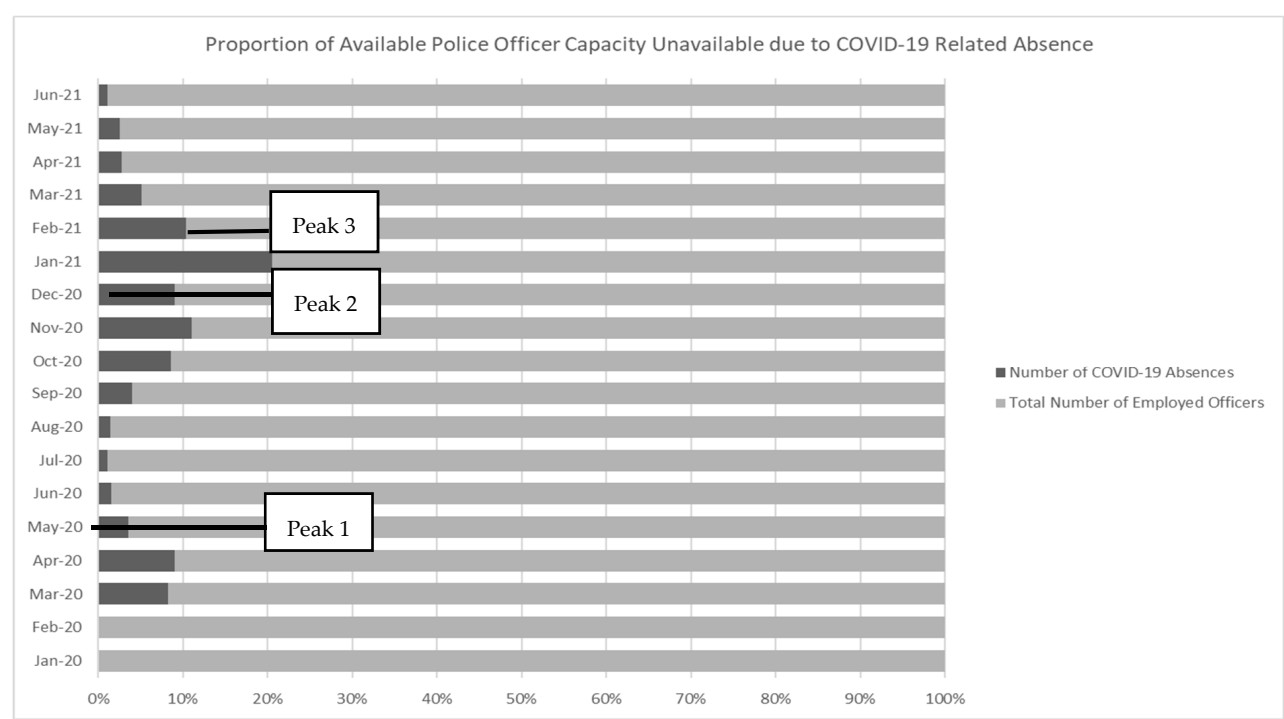

**Figure 1.** Overall impact of COVID-19 on available police officer capacity by month.

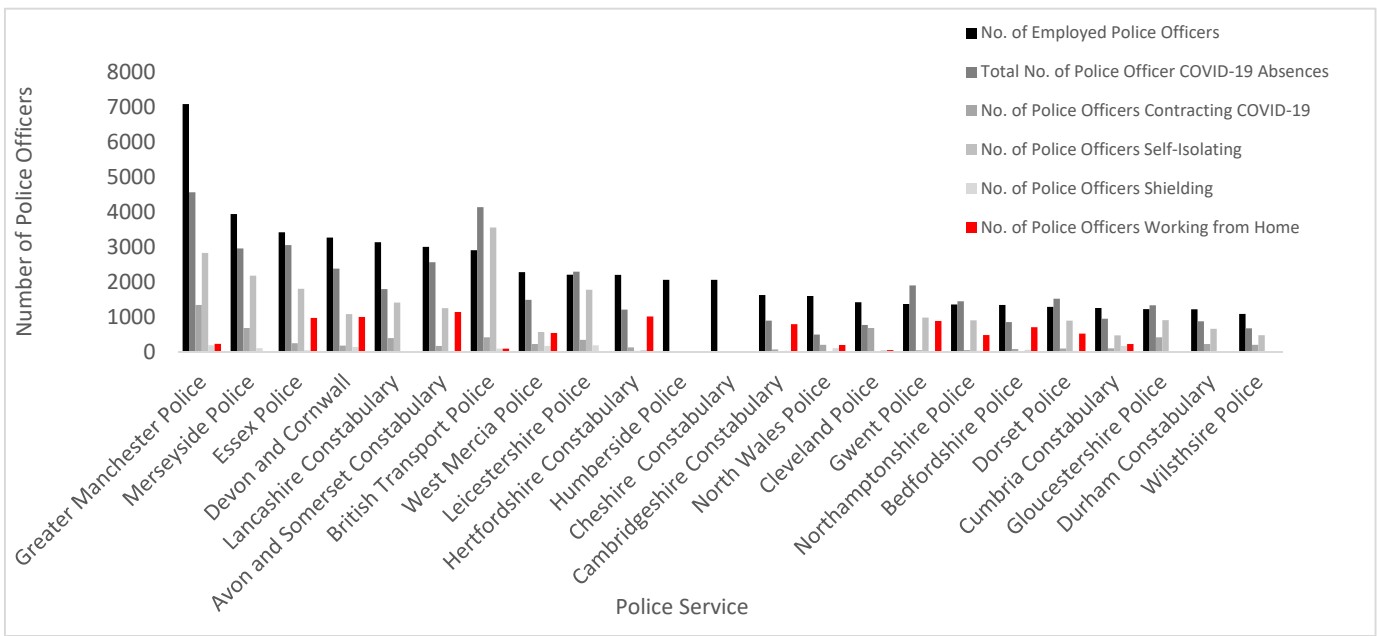

**Figure 2.** Overall impact of COVID-19 on available police officer capacity by police service.

Conversely, smaller forces appeared better equipped to handle this impact through officers conducting work from home, suggesting they were able to respond in a more agile manner. This theme also holds true for police staff members as can be seen in Figure 3. At first glance, an exception to this rule appears to be Police Scotland, which is the second-largest UK force; however, they did not provide data for the number of police staff members who contracted COVID-19 or self-isolated after contact, so this is not a true reflection of their absence levels.

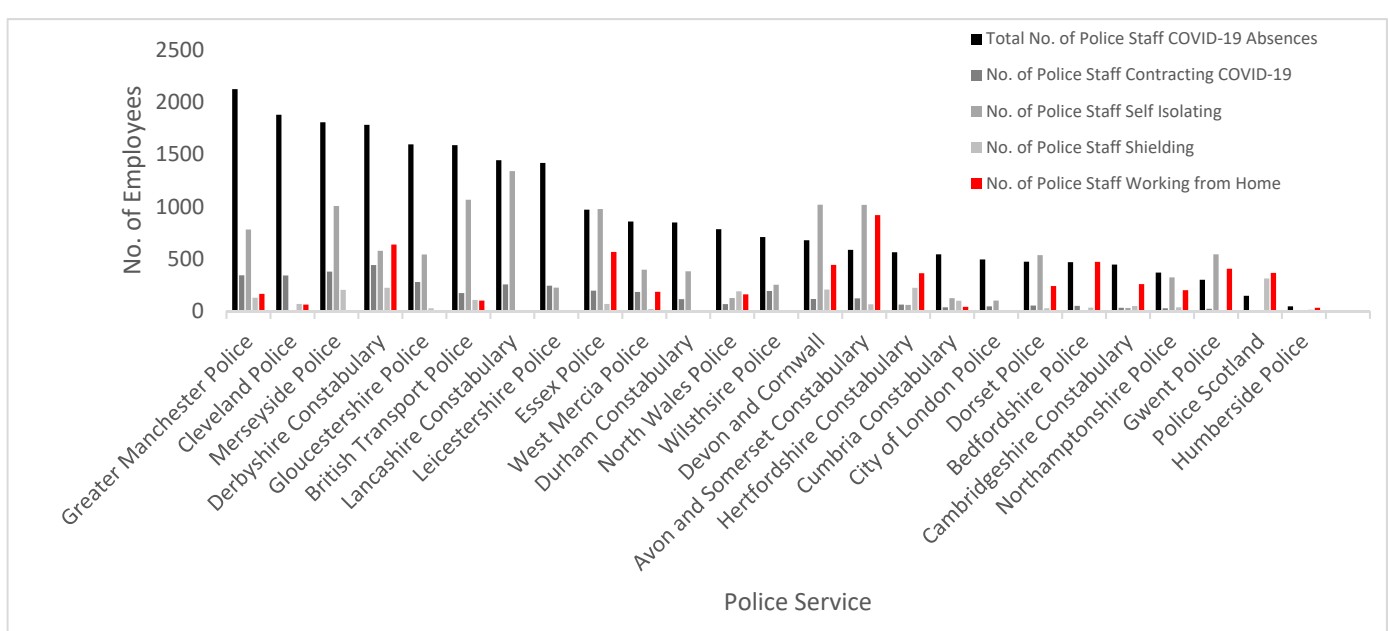

**Figure 3.** Overall impact of COVID-19 on police staff by police service.

In Figures 4 and 5, the trend analysis of overall absences for any of the three reasons follows almost identical trajectories. However, nearly double the number of police officers compared to police staff reported absent due to contracting COVID-19 or because of self-isolation after contact. This is not an unsurprising finding as the overall volume of

warranted police officers in the UK is greater than the number of civilian staff. Given the higher volumes of officers negatively affected by contraction and self-isolation, it would be natural to presume this rule would remain consistent in respect of shielding. However, the number of shielding employees is comparable for both officers and staff. This is likely explainable by differences between the recruitment policies for the different roles. For example, police officers must undergo a rigorous medical examination prior to successful recruitment, and, as such, it is highly probable their rates of serious health conditions that increase vulnerability to COVID-19 are likely to be less. This reduces the necessity for shielding amongst officers, in comparison to police staff, thus making the volume of employees affected more comparable. Shielding also differs in terms of the trends identified.

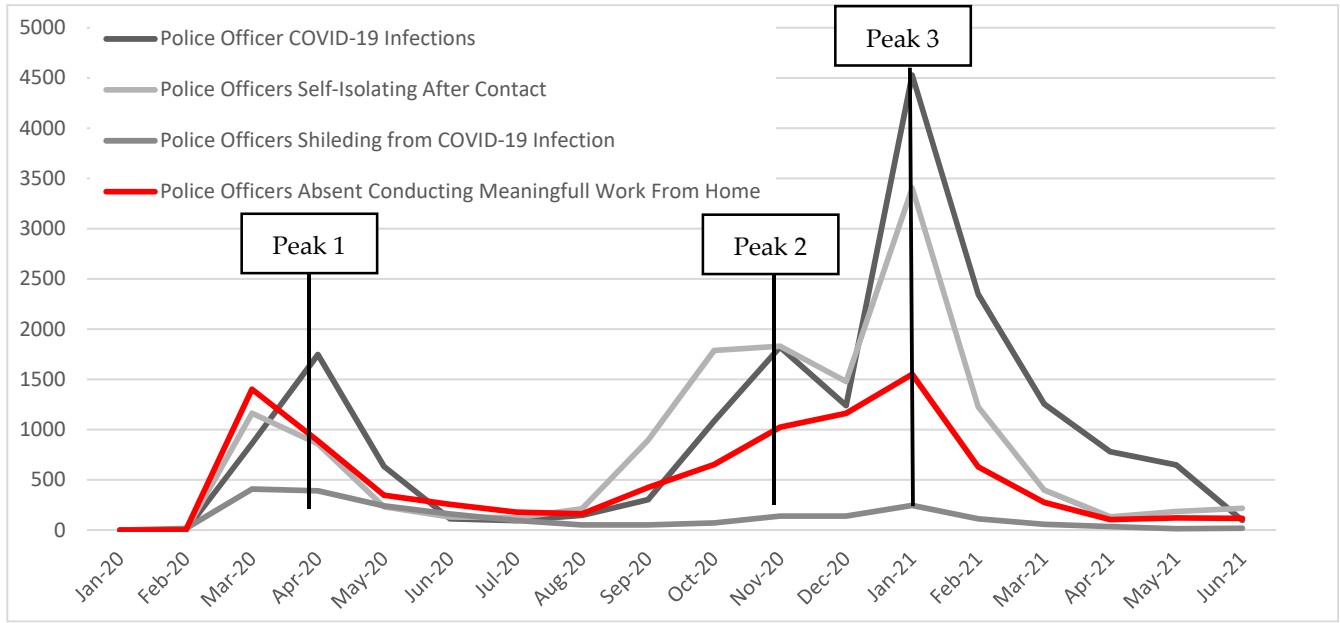

**Figure 4.** Reported police officer absences and peak infection periods vs. police officers conducting meaningful work.

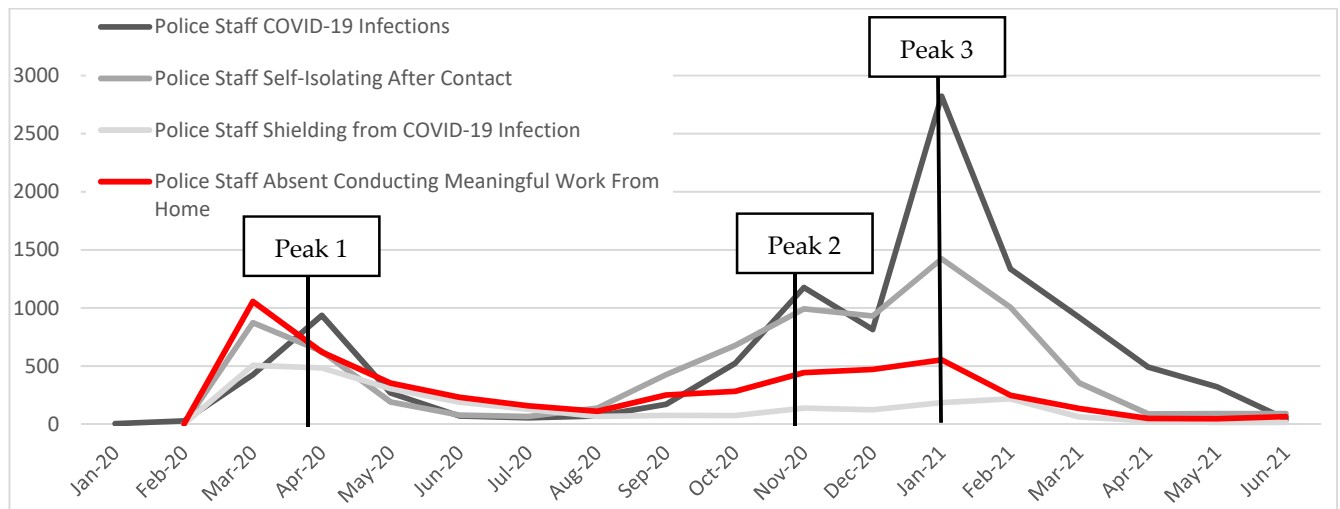

**Figure 5.** Reported police staff absences and peak infection periods vs. Police staff conducting meaningful work.

The beginning of the pandemic is when the levels of shielding for police officers and staff are at their highest, reducing significantly over the summer before rising again when

the infection levels increase. In contrast, the volumes of both officers and staff affected by contraction and self-isolation rise and fall sharply as each wave of the pandemic hits. This is explainable by the consistent nature of shielding. For example, higher numbers of employees were likely to have to shield from COVID-19 at the beginning of the pandemic when knowledge of the risk and threat posed was at its lowest. Over time, as employers adapt, and the rate of vaccination increases, the number of personnel required to shield would likely reduce. In respect of the capability to conduct work from home during any period of absence, greater proportions of police officers (18.9%) were able to do this compared to their police staff colleagues (12.5%). Understanding why is an important issue. One possible reason is the access to more adaptable technology. For example, police officers who conduct frontline duties are far more likely to be personally issued with smart devices and laptops that enable them to work '*in the field' and* operate with greater flexibility.

*7.2. Survey Question Results*

In total, 60 police employees completed the survey from a variety of different departments, which can be seen in Table 2. This is a very small number, and we recommend that future research increase the validity of these early findings by improving the volume of participants. Regardless of its limitations, the analysis of the survey results seen in Table 3 provides further initial insights. In total, 26% (n-16) of respondents to the survey stated that between January 2020 and December 2020 they contracted COVID-19 or had to self-isolate (45%, n-27) as a result of coming into contact with someone who had COVID-19 or who was displaying symptoms. It appears many employees who contracted the virus (50%, n-8) believed that they caught the disease or had to self-isolate (70%, n-19) because of interactions that occurred whilst conducting policing duties. Furthermore, 73% (n-14) of those who declared having to self-isolate had to do so on more than one occasion, with five respondents self-isolating on three or more instances. These findings differ from the FOI data returns in one significant way.

**Table 2.** No. of survey respondents by department.

| Police Department | % of Respondents | No. of Respondents |
|---|---|---|
| 999 responding | 40.68% | 24 |
| Criminal Investigation and Public/Child Protection | 18.64% | 11 |
| Community Policing | 15.25% | 9 |
| Other | 6.78% | 4 |
| Firearms | 5.08% | 3 |
| Force major incident i.e., homicide or counter terrorism | 3.39% | 2 |
| Management of serious or violent offenders | 1.69% | 1 |
| Force control room | 1.69% | 1 |
| Back office i.e., HR/Finance etc. | 1.69% | 1 |
| Traffic | 1.69% | 1 |
| Specialist public order | 1.69% | 1 |
| Digital Investigation | 1.69% | 1 |
| Intelligence, Dog Handling, Mounted and Surveillance or other covert activities | 0.00% | 0 |

**Table 3.** Table of survey responses regarding COVID-19 Infection, self-isolation, and working from home.

| | % of Affected Respondents | No. of Relevant Respondents | No. Answered (Skipped or Not Applicable) |
|---|---|---|---|
| Contracted COVID-19 in 2020 | **27%** | 16 | 60 (0) |
| Believed Contraction Occurred on Duty | **50%** | 16 | 16 (44) |
| Was Required to Self-Isolate in 2020 | **45%** | 27 | 60 (0) |
| Believed the Contact Leading to Self-Isolation Occurred on Duty | **70%** | 19 | 27 (33) |
| Proportion of Those Self-Isolating on 2 or More Occasion's | **73%** | 14 | 27 (33) |
| Proportion of Those Self-Isolating Able to Work from Home | **52%** | 18 | 60 (0) |
| Proportion Required to Shield to Prevent Contracting COVID-19 | **14%** | 8 | 58 (2) |
| Proportion Able to Work from Home Whilst Shielding | **50%** | 4 | 8 (52) |
| Believed they Were Provided Adequate PPE | **47%** | 27 | 58 (2) |
| Believed Inadequate PPE was a Factor in their Infection or Self-Isolation | **18%** | 11 | 60 (0) |
| Believed that Resourcing Capacity was Significantly reduced due to COVID-19 | **83%** | 50 | 60 (0) |

Data supplied by police services indicated that infection by COVID-19 accounted for the highest volume of absences, followed by self-isolation. However, the survey findings indicate self-isolation as the dominant reason for absence. A likely explanation for this is the limited number of respondents (n-60), potentially resulting in unrepresentative findings. As such, further research is required to increase the representation of these results. However, they do begin to offer insights that the FOI data could not achieve alone.

In addition to the insights regarding the impact of COVID-19 on the capacity of the police service, a large proportion (72%, n-43) indicated that they felt there was a significant reduction in access to capabilities during the pandemic. To provide context for these results, Table 4 compares the police departments in which participants indicated had the greatest reductions in capacity vs. those cited as having the largest reductions in specialist capabilities. This analysis indicates that emergency responding (known as immediate response) suffered the greatest reduction in both capacity and associated capabilities. This is important, as these officers are the first responders vital for keeping communities safe and responding to emergency calls for service. Respondents also indicated they felt there was a significant reduction in the access to Taser trained officers (TTO). The capability to carry a taser requires additional training and as such, not all officers do so, meaning any reduction in TTOs can have an impact on the ability of 999 immediate response officers to protect themselves or members of the public. The same can be said of authorized firearms officers. Further examination also identifies that the business areas outlined as being most dramatically affected in terms of reduced capacity do not suffer corresponding reductions in perceived availability for their capabilities. For example, criminal investigation departments are cited as being one of the business areas suffering the greatest reduction in capacity (31%). Correspondingly, these departments contain the overwhelming majority of PIP 2 and PIP 3 accredited investigators. However, the perceived reduction in access to the capability within it was lower (23%).

**Table 4.** Volume of survey respondents who believe access to available capacity or capability was significantly reduced due to COVID-19.

| Capacity | | Capability | |
|---|---|---|---|
| **Policing Department** | **% of Respondents** | | **Specialist Capability** |
| 999 Immediate Response | 80.00% | 60.38% | PIP 1 accredited investigators |
| Community Policing | 43.64% | 28.30% | Community Beat Managers |
| Force control room | 38.18% | 24.53% | Taser trained officer (TTO) |
| Criminal Investigation and Public/Child Protection | 30.91% | 22.64% | PIP 2 and PIP 3 accredited investigators |
| Intelligence | 16.36% | 9.43% | Police analysts |
| Firearms | 14.55% | 16.98% | Authorised firearms officer (AFO) |
| Management of serious or violent offenders | 12.73% | N/A | N/A |
| Roads Policing | 10.91% | 18.87% | Advanced drivers |
| Public order | 10.91% | 20.75% | Public order trained staff (including command courses) |
| | | 9.43% | Specialist search |
| Other | 10.91% | 24.53% | Other |
| Back office i.e., HR/Finance etc. | 10.91% | | |
| Dog handling | 7.27% | 7.55% | Dog Handlers |
| Surveillance of other covert activities | 7.27% | 7.55% | Exhibits officers |
| Digital Investigation | 7.27% | 15.09% | Digital media investigators |
| | | 11.32% | Digital forensic examiners |
| Force major incident i.e., homicide or counter terrorism | 3.64% | 3.77% | Holmes Indexer or other MIR specialisms |
| Mounted | 3.64% | N/A | N/A |

This pattern repeats in respect of emergency response and PIP 1 investigators and community policing and community beat managers (such as PCSOs). These are possibly three of the largest departmental business areas within the police service suggesting the larger the area of business, the higher the reduction in available capacity but the lower the impact of access to capability, presumably because there are simply more of them. In contrast, several areas of business that are perceived to have reduced available capacity appear to have had a disproportionate reduction in access to associated capabilities. For example, the area of digital investigation was only perceived by a low number of respondents (7%) to have reduced available capacity, but the access to the capabilities and skills within that field was higher (11% and 15%, respectively). This is also true for road policing and public order and their associated capabilities.

*7.3. Survey Free Text*

In the final question of the survey, participants were offered the opportunity to express unfiltered views and opinions on the topics covered. Analysis of the free-text responses identified that there was potentially a range of cultural factors that may have influenced

the high volume of recorded absences. For example, a number of employees believed it related to a relaxed attitude to infection exposure:

*"It felt like some colleagues were taking the pandemic much less seriously and had a very laissez-faire attitude to safety. Not only were they not bothered if they caught it (some almost viewing it as a "free holiday/leave"), they weren't bothered about infecting colleagues and the knock on effects that could have had, particularly where family members were higher risk, shielding. Unfortunately making cops less selfish is a tricky one, but perhaps more robust management, earlier on would have helped at least reduce this"*

*"There could/should have been a much more considered approach to resourcing. I felt that in the police we went straight into crisis mode and there was the attitude of 'suck it up', we are all frontline so there will be no working from home! As a result, you had staff crammed into your offices who really did not need to be on duty, particular in back office roles or investigation departments. No surprise then when dozens of staff began having to self-isolate or caught COVID"*

Conversely, others felt that capacity issues were heightened because of overly restrictive isolation policies:

*"As service paid for by the tax payer to protect and support our community and people, police forces should have taken a braver approach and only allowed isolating if people were showing signs. Too many officers have used the COVID stance as a means of having time off"*

*"Less OTT policies for isolating i.e., NHS requirements are if you are fully PPE in a vehicle you don't have to isolate. College of policing have decided you do. So paramedics in the exact same scenario don't isolate but a cop does"*

A number of respondents also cited a lack of effective access to, and use of, infection prevention methods such as the use of personal protective equipment (PPE). In fact, only 47% (n-27) of survey respondents felt that they were provided with adequate PPE and 18% (n-11) believed this was a factor in their necessity to report absent due to COVID-19 infection or self-isolation:

*"More advanced safety over clothing. The present flimsy blue aprons are useless as they only cover the front of your torso. The arms and lower body are still exposed and the kit was issued far too late at the start of the pandemic"*

*"Provide officers and staff with better quality PPE faster than during CV19"*

Understandably, the high proportions of absences had a noticeable impact on available capacity, with 83% (50) of respondents indicating that they believed it was significantly reduced due to COVID-19:

*"There is a general lack of staff with many teams not meeting minimum strengths on a daily basis, added to that COVID and self-isolation it has added much pressure to those who are currently at work"*

*"A greater number of staff is needed to meet the needs of the public without the cost of officers and staff's mental health due to added pressures and work load"*

In respect of the ability to conduct work from home, the survey responses supported the suggestion that employees were unable to rapidly switch to agile working:

*"Main issue was the lack of laptops to allow for agile working outside the office environment"*

*"Understandably, the solutions to this include further investment in information technology solutions"*

*"Invest in sufficient IT equipment to allow officers to do some meaningful work while at home"*

*"All officers/staff at Inspector or equivalent to be provided with a laptop. Afford all staff access to computer systems at home via a confidential portal"*

In addition to providing context to the situation 'on the ground', a number of respondents also provided suggestions to improve future preparedness to mitigate some of the problems experienced:

> *"Keeping teams isolated from other policing teams, too many interact with each other unnecessarily"*

> *"We should have adopted an approach I saw in other agencies with whole departments having in/out weeks and not a mentality to have everyone on duty at a police station just because we are the police. In the end, it really caused huge issues"*

> *"Daily antigen and PCR tests for all frontline officers"*

> *"Provide officers and staff with relevant vaccines or medicines to prevent falling ill"*

> *"Increase reservist pools for specialist departments such as mine Marine Unit"*

> *"Providing more resilience to operational policing"*

## 8. Discussion

As a whole, the study has been successful in answering the questions it set out to and has shown that in general the impact on police demand, capacity, and capability, has been significant, but has not been consistent across the services examined. This suggestion is, of course, caveated in the context of the data collection obstacles identified in the study. The overview of demand helps answer the first question by showing that there have been significant rises in anti-social behavior and organized crime, such as drug-related offences, and cyber-crime. These rises are likely to be compounded by the fact that the police resources used to respond to or investigate these issues are some of the smallest and, or, most specialized. This may explain why the reductions in capacity in specialist functions such as those related to digital investigations, road policing, and public order are felt more acutely in terms of their provision of specialist capabilities. In addition to the rise in crime and disorder, the police also had the additional responsibility of having to implement new coronavirus legislation. It is extremely difficult to understand how much demand this generated but the volume of FPNs issued suggests it created significant work for frontline officers. Given that the approach by police in the UK focused upon the 4 E approach (Engage, Explain, Encourage, Enforce), the number of FPNs is likely to represent only interactions resulting in enforcement and is unlikely to be a true reflection of the full reactive and protective demand placed upon the police service. In the face of this new demand, and compounded by the reductions in available capacity, the decreases in high volume crime types are likely to have been eroded and replaced with new COVID-19 related demand.

As a result, as indicated by the survey results, this exploration study suggests that the ability of the police service to effectively meet its reactive and protective responsibilities may have been jeopardized. Although it could be argued that this highlights a lack of preparedness within policing to respond to the policing requirements of a pandemic, we suggest it further illustrates the broadening remit of the police. Specifically, it highlights further how the scope of policing continues to be widened to include issues of public health, a situation that has been previously identified within law enforcement literature which has outlined the shift of policing into the management of vulnerability (Enang et al. 2019) and mental health (Murray et al. 2018). Importantly, it highlights that the broadening remit of the police continues to impact their ability to respond to their core functions of prevention and reduction of crime and disorder.

In pursuit of its second question, it is very clear that the available capacity suffered large reductions overall and the study indicates that the bulk of the affected resources came from within core policing functions, most notably 999 first responders. That being said, it was not as high as the 40% that some senior officers had reported fearing (Clements and Aitkenhead 2020). The impact on police capacity has followed trends in line with the dates related to peak infection levels of the general population. It also indicates that larger forces appear to have been less able to mitigate reduced capacity through agile working, with

proportionately fewer staff able to work from home. A natural analogy is that it seems to be the difference between turning a tanker versus a car. Ordinarily, the levels of capacity reduction identified in the study should significantly reduce the ability of the police to respond to reactive demand. In theory, this is likely to have been avoided because of the reductions in various crime forms which are most frequently attended by first responders. There was a similar story within the investigation field where demand from crimes such as violent offending, homicide, and sexual offences all reduced in large volumes. However, we suggest any mitigation of reduced capacity through reduced demand is unlikely to have manifested itself. This is because survey respondents suggest that emergency first responding, the investigative field, and community policing were all areas that employees *'felt'* reductions in both capacity and capabilities. This potentially indicates that regardless of reductions in their normative demand, other forms of demand required their attention, such as managing organised crime, investigating cyber-crime, and policing the coronavirus legislation.

Reaching the aforementioned findings helps begin to formulate hypotheses in respect of the likely impact. Emerging research (Clements and Aitkenhead 2020) has indicated that although generally the public believed the police handled the implementation of coronavirus legislation well, there have been negative effects on trust and confidence because of the pandemic. For example, Clements and Aitkenhead (2020) have suggested that members of the public have expressed disappointment at a perceived lack of visibility. Given the rises reported in ASB, this is likely to have negatively affected confidence as research (Jacobson et al. 2005) indicates that police visibility is one of the most effective ways to reassure communities affected by ASB. It has also been suggested that the public felt there was a poorer level of response after reporting issues to the police during the pandemic (Clements and Aitkenhead 2020), although it stops short of concluding the impact of this it is natural to hypothesize that satisfaction may have reduced. We can draw this inference somewhat from the area of cyber-crime. This is an area that has been suggested as a low priority for the police, despite 45% of victims feeling it has a negative impact on their emotional wellbeing (Michael Skidmore and Gill 2020), and, as a result, confidence in the police response is much lower than in other crime areas. With large rises in both cyber-crime and ASB, both areas already experiencing, or being susceptible to, fluctuations in satisfaction, confidence, and trust that relate to visibility and service levels, it is not a stretch to state that these are just two specific areas that the service may have suffered a sizeable reduction in satisfaction, trust, and confidence.

It has been argued that during the pandemic *'policing has been forced to use its discretion on a strategic scale while walking a fine line with the public and with government'* (Clements and Aitkenhead 2020). It is hard to disagree with this statement and be overly critical of the police service response during the pandemic since the scale and scope of the impact were so severe, that it is unlikely any organization got everything right. However, we can begin to consider what lessons can be learnt from this study to improve preparedness for policing during future pandemics. For example, police services could act proactively to rapidly align capacity and capability from areas where demand is likely to reduce, to the areas it is likely to increase, such as cyber and organized crime, ASB, and legislation implementation. In operational terms, this could mean diverting detectives from investigative positions to more specialized functions such as the investigation of cyber dependent, and organised crime. In addition, staff frequently *'ring fenced'* in plain-clothed units, such as those tackling property-related crimes including robbery, burglary, and vehicle crime for example could also be used to target OCGs, or to underpin reductions in available capacity within community policing to tackle rising ASB.

At a strategic level, police services can also ensure they are better prepared by identifying areas that are high capability and low capacity, such as digital investigation, and ensuring the service has a plan for mitigation of the effects in those areas. This could include increasing capacity by retaining reserve specialists, volunteers, or special constables, raising the knowledge and capability of the wider workforce, or altering submission or

deployment thresholds. It was also apparent from the study that many police services appear digitally unequipped to shift to agile working and enable high proportions of absent staff to work from home. To address this issue, the police service could hasten its efforts further to continue the digitization of frontline police services. As a minimum, they could retain a list of all employees who have serious health conditions that are most susceptible to serious illnesses. Accepting that each pandemic may be different in cause and effect, it is natural to presume that employees vulnerable to COVID-19 may be more susceptible to other serious diseases. To mitigate the impact of shielding in future pandemics, vulnerable employees could be prioritized for technology that enables agile working so that should the situation repeat, they can rapidly switch to working from home with minimum impact on overall service delivery. To reduce the strategic impact of the reduced capacity of all employees, police services should consider securing stockpiles of PPE to avoid the early impact of a lack of access to such safeguarding measures. In addition, contingency plans for the workplace could be improved upon so that they enable employees to socially distance more effectively. Solutions such as prioritization of core frontline and back-office functions, rapid changes to shift patterns to reduce crossover of staff, plans to maximize estate, and investing in more agile ICT infrastructure will all aid this process. For example, traditional desktop personal computers have little value in a pandemic environment, and the roll-out of personal-issued laptops with docking stations would be a much more sensible solution. Having the correct equipment for staff to work in an agile fashion would have a multiplier effect by enabling greater flexibility in work location and deployment of staff. This could contribute to reducing the likelihood of infection transmission, contact, and subsequent self-isolation as a result of workplace-related contact, which was a major issue cited by survey respondents.

## 9. Limitations

Although this study has made some progress in understanding how the changing demand for police services in the UK has impacted both their capacity and capability, there are a number of limitations. First, the response rate to the FOI requests was only 58%. As a result, it is not possible to conclusively identify the overall impact on the capacity and capability of the police service in the UK. To achieve this, data from every service would be required and at this time there is no way to obtain this beyond the FOI methodology without the support of key stakeholders within UK policing. Similarly, the survey response rate was very limited and although it provides some early insights, any generalizations should be considered with caution. To improve the validity of the findings in both areas, we recommend a nationwide study of UK police services to explore the issues raised further. This would be best achieved in conjunction with the National Police Chiefs council or the College of policing.

## 10. Conclusions

This study sought to understand the impact of COVID-19 on police demand, capacity, and capability, and the effect of this on the police. It also sought to provide potential solutions for future preparedness. The study has partially achieved these aims and begun to identify that demand shifted in predictable, but also unanticipated ways. There was also a substantial impact on both the capacity and capability of the police service in core functions such as 999 responding, criminal investigation, and community policing. In terms of capability, the impact of COVID-19 was acutely felt in departments that were low capacity, and high capability. The effects of these findings are discussed in terms of their possible impact on the satisfaction, trust, and confidence of the police and it is suggested that this is likely to have been negatively affected during the pandemic due to the reduced ability of the police to meet the demands placed upon them. Any drop in satisfaction, trust, and confidence is likely to have been most prominent in victims of cyber-crime and anti-social behavior, both areas that suffered significant rises during the pandemic. In an effort to be better prepared for future similar policing climates, a number

of solutions were proposed including continued investment in more agile ICT and careful review and monitoring of contingency plans that can be used to more rapidly shift to a changing environment, most notably the risk presented for future policing pandemics.

**Funding:** This research received no external funding.

**Institutional Review Board Statement:** The study was conducted in accordance with the Declaration of Helsinki, and approved by the Ethics Committee of Rabdan Academy, Al Dhafeer Street, Abu Dhabi, United Arab Emirates under application reference #0006 on November 23rd 2021.

**Informed Consent Statement:** Informed consent was obtained from all subjects involved in the study.

**Data Availability Statement:** Data for this study is retained by the authors affiliated university. This can be requested by direct contact with the author.

**Conflicts of Interest:** The authors declare no conflict of interest.

## Appendix A

**Table A1.** Percentile Impact Identified from Studies of the Impact of COVID-19 on UK Police Reactive Demand (Recorded Crime and Disorder).

| Crime Type | Geographic Area | Impact | Additional Information | Data/Lockdown Period Examined | Source |
|---|---|---|---|---|---|
| Theft from the Person | England and Wales | Decreased by 79.2% | Only examined 1 month during lockdown | April 2020 | (Dixon et al. 2020) |
| | England | Decreased by 77.6% Decreased by 44.4% | During national lockdown After national lockdown | March 2020–May 2021 | (Neanidis and Rana 2021) |
| Shoplifting | England and Wales | Decreased by 36% | Crime Survey of England and Wales | March 2020–March 2021 | (CSEW 2021) |
| | England and Wales | Decreased by 55.9% | Only examined 1 month during lockdown | April 2020 | (Dixon et al. 2020) |
| | Lancashire | Decreased by 61.6% | Only examined 1 week after lockdown | 23rd March–29th March 2020 | (Halford et al. 2020) |
| Robbery | United Kingdom | Decreased by 60% | Gradual increase over 6 months but remained significantly lower | March 2020–August 2020 | (Langton et al. 2020) |
| | England and Wales | Decreased by 34% | Crime Survey of England and Wales | March 2020–March 2021 | (CSEW 2021) |
| | England and Wales | Decreased by 57.6% | Only examined 1 month during lockdown | April 2020 | (Dixon et al. 2020) |
| | England | Decreased by 52% Decreased by 32.6% | During national lockdown After national lockdown | March 2020–May 2021 | (Neanidis and Rana 2021) |
| | London | Decreased by 54% | Reductions are on daily counts | 1 January 2020–30 April 2020 | (Nivette et al. 2021) |

**Table A1.** *Cont.*

| Crime Type | Geographic Area | Impact | Additional Information | Data/Lockdown Period Examined | Source |
|---|---|---|---|---|---|
| Domestic Abuse | England and Wales | Increased by 6% | Crime Survey of England and Wales | March 2020–March 2021 | (CSEW 2021) |
| | Lancashire | Decreased by 44.7% | Reduced citizen mobility | 23 March–29 March 2020 | (Halford et al. 2020) |
| Burglary | United Kingdom | Decreased by 20% | Gradual increase over 6 months but remained significantly lower by 10% | March 2020–August 2020 | (Langton et al. 2020) |
| | England and Wales | Decreased by 37.1% | Only examined 1 month during lockdown | April 2020 | (Dixon et al. 2020) |
| | England and Wales | Decreased by 30% | Crime Survey of England and Wales | March 2020–March 2021 | (CSEW 2021) |
| | England | Decreased by 24.3% Decreased by 19% | During national lockdown After national lockdown | March 2020–May 2021 | (Neanidis and Rana 2021) |
| | London | Decreased by 41.6% | Reductions are on daily counts | 1 January 2020–30 April 2020 | (Nivette et al. 2021) |
| | Lancashire | Non-dwelling decreased by 25.6%. Dwelling reduced by 25.4% | Only examined 1 week after lockdown | 23 March–29 March 2020 | (Halford et al. 2020) |
| Vehicle Theft | England and Wales | Decreased by 28% | Crime Survey of England and Wales | March 2020–March 2021 | (CSEW 2021) |
| | England and Wales | Decreased by 41.2% | Only examined 1 month during lockdown | April 2020 | (Dixon et al. 2020) |
| | England | Decreased by 36.8% Decreased by 30.9% | During national lockdown After national lockdown | March 2020–May 2021 | (Neanidis and Rana 2021) |
| | London | Decreased by 30.7% | Reductions are on daily counts | 1 January 2020–30 April 2020 | (Nivette et al. 2021) |
| | Lancashire | Theft of increased by 1.1%. Theft from decreased by 43.3% | Reduced citizen mobility | 23 March–29 March 2020 | (Halford et al. 2020) |

**Table A1.** *Cont.*

| Crime Type | Geographic Area | Impact | Additional Information | Data/Lockdown Period Examined | Source |
|---|---|---|---|---|---|
| Other Theft | United Kingdom | Decreased by 80% | Gradual increase over 6 months but remained significantly lower | March 2020–August 2020 | (Langton et al. 2020) |
| | England and Wales | Decreased by 32% | Crime Survey of England and Wales | March 2020–March 2021 | (CSEW 2021) |
| | England | Decreased by 36% Decreased by 24.4% | During national lockdown After national lockdown | March 2020–May 2021 | (Neanidis and Rana 2021) |
| | London | Decreased by 54.4% | Reductions are on daily counts | 1 January 2020–30 April 2020 | (Nivette et al. 2021) |
| | Lancashire | Decreased by 52.4% | Reduced citizen mobility | 23 March–29 March 2020 | (Halford et al. 2020) |
| Assaults | England and Wales | Decreased by 28% | Crime Survey of England and Wales | March 2020–March 2021 | (CSEW 2021) |
| | London | Decreased by 12.3% | Reductions are on daily counts | 1 January 2020–30 April 2020 | (Nivette et al. 2021) |
| | Lancashire UK | Decreased by 35.6% | Reduced citizen mobility | 23 March–29 March 2020 | (Halford et al. 2020) |
| Homicide | England and Wales | Decreased by 16% | Crime Survey of England and Wales | March 2020–March 2021 | (CSEW 2021) |
| | London | Decreased by 25% | Reductions are on daily counts | 1 January 2020–30 April 2020 | (Nivette et al. 2021) |
| Public Order | United Kingdom | Decreased by 20% | Quickly increases and within 2 months returns to pre-COVID levels | March 2020–August 2020 | (Langton et al. 2020) |
| | England and Wales | Decreased by 17.3% | Only examined 1 month during lockdown | April 2020 | (Dixon et al. 2020) |
| Sexual Violence | United Kingdom | Decreased by 24% | Gradual increase to pre-COVID levels over 6 months | March 2020–August 2020 | (Langton et al. 2020) |
| | England | Decreased by 19% Decreased by 4.3% | During national lockdown After national lockdown | March 2020–May 2021 | (Neanidis and Rana 2021) |
| Criminal Damage | United Kingdom | Decreased by 20% | Gradual increase over 6 months to pre-COVID levels | March 2020–August 2020 | (Langton et al. 2020) |
| | England and Wales | Decreased by 30.1% | Only examined 1 month during lockdown | April 2020 | (Dixon et al. 2020) |
| | England | Decreased by 20.3% Decreased by 6.8% | During national lockdown After national lockdown | March 2020–May 2021 | (Neanidis and Rana 2021) |

**Table A1.** *Cont.*

| Crime Type | Geographic Area | Impact | Additional Information | Data/Lockdown Period Examined | Source |
|---|---|---|---|---|---|
| Possession of Offensive Weapons | England and Wales | Decreased by 8.8% | Only examined 1 month during lockdown | April 2020 | (Dixon et al. 2020) |
| | England | Decreased by 10.5% | During national lockdown | March 2020–May 2021 | (Neanidis and Rana 2021) |
| Organised Crime (Inc. Drug Trafficking/Possession) | United Kingdom | Increased by 30% | Rapid after 2 months to statistically reduced level of 10% | March 2020–August 2020 | (Langton et al. 2020) |
| | England and Wales | Increased by 9.8% | Only examined 1 month during lockdown | April 2020 | (Dixon et al. 2020) |
| | England | Increased by 28.5% Increased by 8.6% | During national lockdown After national lockdown | March 2020–May 2021 | (Neanidis and Rana 2021) |
| Cyber Crime | United Kingdom | Increased by 43.24% | Only includes cyber dependent crime and online fraud | May 2020 | (Buil-Gil et al. 2020) |
| | England and Wales | Increased by 28% | Crime Survey of England and Wales | March 2020–March 2021 | (CSEW 2021) |
| ASB | United Kingdom | Increased by 100% | Rapid after 2 months to statistically reduced level of 10% | March 2020–August 2020 | (Langton et al. 2020) |
| | England and Wales | Decreased by 108.9% | Only examined 1 month during lockdown | April 2020 | (Dixon et al. 2020) |
| | England and Wales | Increased by 28% | Crime Survey of England and Wales | March 2020–March 2021 | (CSEW 2021) |
| | England | Increased by 65.5% Increased by 22.9% | During national lockdown After national lockdown | March 2020–May 2021 | (Neanidis and Rana 2021) |
| Breaches of Coronavirus (COVID-19) Legislation | England and Wales | Comparison not possible | 117,213 individual fixed penalty fines issued by Police | March 2020–20th June 2021 | (NPCC 2021) |

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
