# Peer review of "An Exploration of the Impact of COVID-19 on Police Demand, Capacity and Capability"

_socsci, doi:10.3390/socsci11070305_

Round 1

Reviewer 1 Report

Thank you for the opportunity to review this paper. It was a pleasure to read, and overall, I find that it is a good contribution to policing literature. I would like to thank the authors for their work on an issue of importance for the field.

The methodology is appropriate and described in detail. Thank you.

I particularly like the timing of this article, at a time when police (and policing scholars) have to 'take stock' of what has been happening. The paper is well written, and in a (justifiably) assertive manner: I liked, for example, statements such as 'the ability of the police service to effectively meet its reactive and protective responsibilities may have been jeopardised'. This is true, and the authors have the evidence to make such incisive comments. Well done. However, it would be better to have these statements better framed within stronger (or at least better sign-posted) and earlier definitions of 'capacity and capability' v. what is expected within the 'demand'. Demand is very well explored in the first § of the literature review section, but capability and capacity, less so. These terms are only briefly touched upon in the introduction and first pages of the paper. I think some specifics are needed early on in the paper, to premise what comes in the rest of the article.

It is only when I arrived to the middle of page 19 (2nd §) that I thought there really needed to be some kind of an all-wrapping, overarching statement about the constantly expanding remit of policing and the articulations between law enforcement and public health (see LEPH literature) and the state of preparedness of police within the circumstances of pandemics and their broader, more technological portfolio. I think this was the only 'glaring' gap in this paper, which overall (I repeat), I really liked reading.

With my best wishes to the authors. I am looking forward to seeing this article in print

Some editorial issues / questions:

The first § of section 2 (page 2) is quite heavily referenced (only refs over 6 lines). I understand that this is part of the lit rev of the article - but is this necessary?

Page 2: 'failure demand'. I think this needs a more specific and direct definition, as I think some readers unfamiliar with the expression might think the authors mean 'demand failure'.

The table on pages 6/7 is very interesting, but difficult to read an,d overall grasp. Please consider reformatting and styling it differently so readers can have it at a glance. Some suggestions: reduce space around text to reduce white space, use arrows (in lieu of the repetition of increase / decrease) to reduce text, are some categories 'groupable' under a more generic heading/definition?

Page 12, lines 355-359: sentence too long. Break down.

Author Response

Reviewer 1

(Capability and capacity) These terms are only briefly touched upon in the introduction and first pages of the paper. I think some specifics are needed early on in the paper, to premise what comes in the rest of the article.

To address this feedback, I have now added several sentences to introduce these components during the introduction section as follows;

“Vital to their response is also the understanding of the level of available capacity, as defined by the numbers of police officers and civilian police staff, as any failure to adequately match incoming demand with the requisite level of capacity is likely to result in significant impact on service delivery. In addition, capability, as defined by the specialist skills and training that certain police officers and civilian staff possess, is also key as certain forms of incoming demand can only be managed by officers or staff with specific qualifications. When considering these factors, we can begin to appreciate how each has a direct and significant influence over the other”

There really needed to be some kind of an all-wrapping, overarching statement about the constantly expanding remit of policing and the articulations between law enforcement and public health (see LEPH literature) and the state of preparedness of police within the circumstances of pandemics and their broader, more technological portfolio. I think this was the only 'glaring' gap in this paper, which overall (I repeat), I really liked reading.

I have added a short paragraph to the discussion section that now captures these views, supported by two chosen pieces of existing literature, as follows;

“Although it could be argued that this highlights a lack of preparedness within policing to respond to the policing requirements of a pandemic, we suggest it further illustrates the broadening remit of the police. Specifically, it highlights further how the scope of policing continues to be widened to include issues of public health, a situation that has been previously identified within law enforcement literature which has outlined the shift of policing into the management of vulnerability (Enang et al, 2019) and mental health (Murray et al, 2018). Importantly, it highlights that the broadening remit of the police continues to impact upon their ability to respond to their core functions of prevention and reduction of crime and disorder”

'Failure demand', I think this needs a more specific and direct definition, as I think some readers unfamiliar with the expression might think the authors mean 'demand failure'.

I have further defined the definition of failure demand to include the following as requested;

“Failure demand also includes work conducted that does not provide the customer, which in a policing context would generally be the victim of crime, an outcome that provided value to them (Benington and Moore, 2011).

The table on pages 6/7 is very interesting, but difficult to read and overall grasp. Please consider reformatting and styling it differently so readers can have it at a glance. Some suggestions: reduce space around text to reduce white space, use arrows (in lieu of the repetition of increase / decrease) to reduce text, are some categories 'groupable' under a more generic heading/definition?

To make the table more manageable I have been able to reduce the white space as advised, and increased the font size by 1. I have also moved it out of the main text and into an appendix which I feel serves to further remove confusion and is the most appropriate positon for such a comprehensive table.

Of note, the data is already categorised based on the crime classifications. This is the most natural way to categorise given the differing results the studies provided.

Page 12, lines 355-359: sentence too long. Break down.

This has been shortened appropriately

Reviewer 2 Report

Thanks for the opportunity to review this paper.

In the second sentence of the abstract, the authors write that focusing on the UK enables the authors to draw conclusions with global implications. I did not understand why results from the UK are so generalizable.

The first paragraph should be split up into at least three separate paragraphs. There are overly long paragraphs throughout the paper.

The methodology section lists a literature review as the first step. A literature review of this type is not a methodology (systematic reviews and meta-analyses are methodologies, but typical literature reviews are not). The description of the search terms and selection criteria is confusing. Are the authors attempting to do a systematic review here?

The paper tries to do too much. The literature review in the findings section is confusing, particularly since there is already a literature review in section 2 of the paper. The response rate to the freedom of information requests was weak and the findings not particularly informative. The methodology for the survey is not clearly articulated. This reads like three papers merged into one.

I encourage the authors to split this into multiple papers and do a much more thorough job in each one of articulating the methodology fully. If they are seeking to do a systematic review (which is not clear), then they should adopt the terminology and typical structure of a published systematic review.

The writing could also be more polished. The authors may want to have their work copyedited before submitting it for publication.

This is important work. I wish the authors well in their efforts to get it published.

Author Response

Reviewer 2

In the second sentence of the abstract, the authors write that focusing on the UK enables the authors to draw conclusions with global implications. I did not understand why results from the UK are so generalizable

This has been removed to improve focus of the abstract.

The first paragraph should be split up into at least three separate paragraphs. There are overly long paragraphs throughout the paper.

This has been split as recommended at appropriate intervals.

The methodology section lists a literature review as the first step. A literature review of this type is not a methodology (systematic reviews and meta-analyses are methodologies, but typical literature reviews are not). The description of the search terms and selection criteria is confusing. Are the authors attempting to do a systematic review here?

I agree, in hindsight, this section was somewhat confusing. I have simplified the method section by merging the reviews of literature from the introductory section and the one from the findings section, into the literature review. The literatire review parameters are not necessary as the paper does not seek to conduct a systematic review, as such, these have been removed.

The result is a much simpler and fluid review and removal of the issues surrounding the examination of literature as a methodology.

The literature review in the findings section is confusing, particularly since there is already a literature review in section 2 of the paper. 

The response rate to the freedom of information requests was weak and the findings not particularly informative.

I respect the reviewer comments on this aspect but would like to point out that overall the response rate was 58%. Considering the sensitivity of the data requested from the police services I believe this is very high. It is not unusual for them to simply refuse to disclose this, as a number did in this case, so any results obtained provide valuable information for the policing scholarly community.

In respect of the findings, I firmly believe they are important (as supported by reviewer 1 and 3). My advice would be that to fully appreciate them the findings have to be considered from a practitioner perspective. When this is done one can more clearly appreciate them in the context of strategically and tactically managing a large organization, such as a police service.

The methodology for the survey is not clearly articulated.

Further detail has been added that outlines the survey method for question generation, survey hosting, circulation, dissemination and analysis as follows;

To create the survey 19 questions were split into 4 key categories. These included; (1) demographic information, such as length of service and age (2) capability, which included details of the participant’s present role, any specialist qualifications they held (3) capacity, such as periods of sickness or absence due to COVID-19 and (4) COVID related information which explored the circumstances of the diseases impact on the participant’s ability to conduct their role, or other work effectively. The survey questions were then uploaded to a third party web hosting service which enabled digital completion of the survey by participants who could do so by following a link provided. The survey was voluntary and this was declared at the outset.

And

To display the findings descriptive approaches are utilised as amongst the primary target audiences for this article are practitioners, senior police leaders and policy makers. It has been suggested that the use of descriptive approaches are more effective at improving understanding (Conner and Johnson, 2017), and therefore have a greater potential for impact.

The writing could also be more polished. The authors may want to have their work copyedited before submitting it for publication.

I am a native English speaker and writer and as such have not used the services of a copyeditor but I have completed a further proof read, along with several other academic colleagues, and all errors appear to have been rectified.

I encourage the authors to split this into multiple papers and do a much more thorough job in each one of articulating the methodology fully.

There are only 2 methods now used in this paper. In my view, alone, neither would provide the depth and richness that combining both achieves. This was my intention and I think that the triangulation of the data compliments each other well. I would strongly recommend they are kept as a single paper because of these reasons and I would respectfully urge you to take into account the glowing comments from reviewers 1 and 2 on this matter.

If they are seeking to do a systematic review (which is not clear), then they should adopt the terminology and typical structure of a published systematic review.

I have amended the literature review section and it should now be clear that the paper is not a systematic review.

Reviewer 3 Report

This is a well written and structured paper that addresses some interesting and valid points. The use of the tree different methods gives a depth of empirical knowledge and information that provides the reader with a clear picture of the issues at hand. The presentation of graphs and tables is good and there is a detailed discussion of points raised. The addition of qualitative data is also good. A worthy addition o the field of study.

Author Response

Reviewer 3

There are no necessary elements identified by the reviewer for amendment

Not applicable

Round 2

Reviewer 2 Report

The authors have addressed some of my concerns. The paper still requires an additional round of copy editing. The writing is still somewhat unpolished.

Author Response

Reviewer 2 has stated that the “findings not particularly informative”. I would respectfully disagree and would urge you not to be de-swayed from publication by this comment. As a former senior police officer I would have found significant value in these findings when conducting strategic planning. This is a position also echoed by both reviewer 1 and 3 who gave glowing praise for the paper and as such, I believe the publication would be very well received, particularly amongst practitioners in the policing and security community.
Second, reviewer 2 has also suggested they “encourage the authors to split this into multiple papers and do a much more thorough job in each one of articulating the methodology fully”. I believe much of the original confusion was caused by the initial inclusion of a literature review as a methodology. I have now addressed this and have also made improvements to the survey method section that I believe meet the required standard.